# Sexually Dimorphic Effects of Histamine Degradation by Enteric Glial Histamine *N*-Methyltransferase (HNMT) on Visceral Hypersensitivity

**DOI:** 10.3390/biom13111651

**Published:** 2023-11-14

**Authors:** Jonathon L. McClain, Wilmarie Morales-Soto, Jacques Gonzales, Visha Parmar, Elena Y. Demireva, Brian D. Gulbransen

**Affiliations:** 1Department of Physiology, Michigan State University, 567 Wilson Road, East Lansing, MI 48824, USA; mcclai11@msu.edu (J.L.M.); morales-soto.wilmarie@mayo.edu (W.M.-S.); gonza984@msu.edu (J.G.);; 2Transgenic and Genome Editing Facility, Institute for Quantitative Health and Engineering, Michigan State University, 567 Wilson Road, East Lansing, MI 48824, USA; demireva@msu.edu

**Keywords:** histamine *N*-methyltransferase (HNMT), histamine, enteric glia, enteric nervous system, intestine, visceral pain, gut, autonomic

## Abstract

Histamine is a neuromodulator that affects gut motility and visceral sensitivity through intrinsic and extrinsic neural pathways, yet the mechanisms regulating histamine availability in these pathways remain poorly understood. Here, we show that enteric glia contribute to histamine clearance in the enteric nervous system (ENS) through their expression of the enzyme histamine *N*-methyltransferase (HNMT). Glial HNMT expression was initially assessed using immunolabeling and gene expression, and functionally tested using CRISPR-Cas9 to create a Cre-dependent conditional *Hnmt* ablation model targeting glia. Immunolabeling, calcium imaging, and visceromotor reflex recordings were used to assess the effects on ENS structure and visceral hypersensitivity. Immunolabeling and gene expression data show that enteric neurons and glia express HNMT. Deleting *Hnmt* in Sox10+ enteric glia increased glial histamine levels and altered visceromotor responses to colorectal distension in male mice, with no effect in females. Interestingly, deleting glial *Hnmt* protected males from histamine-driven visceral hypersensitivity. These data uncover a significant role for glial HNMT in histamine degradation in the gut, which impacts histamine-driven visceral hypersensitivity in a sex-dependent manner. Changes in the capacity of glia to clear histamines could play a role in the susceptibility to developing visceral pain in disorders of the gut–brain interaction.

## 1. Introduction

Disorders of the gut–brain interaction (DGBI) are common [1] and often accompanied by symptoms that include altered mucosal immune function, dysmotility, and visceral pain [2,3,4]. Neuroplasticity, microbiota alterations, immune activation, and abnormal communication along the gut–brain axis are among the factors that contribute to DGBI pathophysiology, but the underlying mechanisms remain incompletely understood. Neuroplasticity, in particular, plays a major role in producing persistent changes in visceral sensations and motor function in irritable bowel syndrome (IBS), an exemplary DGBI [5,6]. Changes in the sensitivity of enteric neurons disrupt the synaptic communication of myenteric neurocircuits that control gut motility, while increased sensitivity of gut-innervating neurons in dorsal root ganglia contributes to heightened perceptions of visceral pain.

Several candidate molecules have been proposed to contribute to enteric and sensory neuroplasticity; however, histamines have emerged as particularly important mediators in contexts that produce visceral hypersensitivity [7,8,9]. Histamine levels are elevated in individuals with diarrhea-predominant IBS [10] and correlate with heightened sensations of pain [11] involving causative mechanisms of action on H1 and H4 histamine receptors. H1 histamine receptors are expressed by sensory neurons, and activation subsequently facilitates signaling through transient receptor potential (Trp) vanilloid 4 (Trpv4) [12], Trp ankyrin 1 (Trpa1), and Trpv1 channels [13]. Elevations in histamine at the level of the intestinal mucosa also contribute to the activation of submucosal neurons in the ENS [14]. Drugs targeting the histamine pathway, such as ebastin, a histamine type 1 (H1) receptor antagonist [13], and cyproheptadine, an antihistamine with anticholinergic and antiserotonergic properties [15], have beneficial effects on visceral hypersensitivity. H4 histamine receptors are expressed by numerous cells in the colon, including epithelial cells, mast cells, goblet cells, intraepithelial lymphocytes, macrophages, and enteroendocrine cells [11]. The activation of H4 receptors is involved in the recruitment of mast cells [11,16] and H4 receptor antagonism reduces visceral pain [11,16,17,18].

Elevations in mucosal histamine during disease can occur through increases in production and/or defects in degradation. Sources of intestinal histamine production are myriad and include immune cells [19,20], epithelial cells [21,22], and neurons [23] in the gut wall, while the largest source of histamine potentially comes from the microbiota [11,24]. Increased histamine production by mast cells and the microbiota are both considered important to the pathophysiology of visceral pain [7,11]. In contrast, how changes in histamine degradation contribute to visceral pain and what cells are involved is less clear. Histamine is metabolized almost exclusively through one of two enzymatic pathways: removal via oxidation occurs through the actions of diamine oxidase (DAO), while degradation via methylation occurs via histamine *N*-methyltransferase (HNMT). DAO is expressed in the intestinal epithelium [25] and contributes to metabolizing histamine contained in food, and DAO deficiency is implicated in histamine intolerance [26,27]. HNMT plays a larger role in metabolizing histamine in the nervous system and is the primary mechanism responsible for controlling histamine in the brain [28,29,30,31]. Astrocytes are the main cellular site of HNMT expression in the brain and are responsible for the bulk of histamine re-uptake and degradation [32,33].

In prior work, we found that enteric glia, a type of peripheral neuroglia functionally related to astrocytes in the brain, detect extracellular histamine and have the potential to concentrate intracellular histamine [34]. Based on the known role of astrocytes in degrading histamine through HNMT, we hypothesized that enteric glia express HNMT, and that glial histamine metabolism regulates visceral hypersensitivity. We tested our hypothesis by assessing HNMT expression in the ENS and testing its functional role by creating a glial-specific conditional *Hnmt* ablation model. The in vivo measures and ex vivo recordings of nociceptive nerve terminals in mice expressing genetically encoded Ca^2+^ indicators were used to study the effects on visceral sensitivity. The data show that while both enteric glia and neurons have the potential to express the histamine degradation enzyme HNMT, the targeted deletion of glial *Hnmt* produces changes in intracellular histamine content and visceral sensitivity in a sex-dependent manner. Therefore, glia-mediated histamine metabolism via HNMT contributes to factors modifying neuroplasticity in the gut and may contribute to changes involved in the DGBI.

## 2. Materials and Methods

### 2.1. Animals

All experimental protocols were approved on 7/7/2021 for animal protocol 202100150 by the Michigan State University Institutional Animal Care and Use Committee (IACUC) in specific pathogen-free conditions in facilities accredited by the Association for Assessment and Accreditation for Laboratory Care (AAALAC) International. Mice of both sexes were used for experiments at 12–18 weeks of age. The mice were maintained in a temperature-controlled environment (Optimice^®^ cage system; Animal Care Systems Inc., Centennial, CO, USA) on a 12 h light: dark cycle with access to water and a minimal phytoestrogen diet (Diet Number 2919; Envigo, Indianapolis, IN, USA) ad libitum.

Transgenic mice with a tamoxifen-sensitive knockout of *Hnmt* in enteric glia (*Sox10^CreERT2^*;*Hnmt*) were bred in-house and generated by crossing *Sox10^CreERT2+/−^* [35] (MGI:5910373; a gift from Dr. Vassilis Pachnis, The Francis Crick Institute, London, UK) with mice containing loxP sites flanking exon 3 of the *Hnmt* gene (generated by Michigan State University’s Transgenic and Genome Editing Facility) to generate *Sox10^CreERT2+/−^;Hnmt^f/f^* mice. Mice with a global deletion of *Hnmt* (*CMV^Cre^*;*Hnmt*) were bred in-house and generated by crossing *CMV^Cre+/−^* [36] [B6.C-Tg(CMV-Cre)1Cgn/J; Jackson Laboratory, Bay Harbor, ME, stock number 006054; RRID: IMSR_JAX:006054] with *Hnmt flox* mice to generate *CMV^Cre+^;Hnmt^f/f^* and *CMV^Cre−^;Hnmt^f/∆^* mice. Transgenic mice expressing the genetically encoded Ca^2+^ indicator GCaMP5g in sensory neurons (*TRPV1^Cre^;GCaMP5g-tdT* mice) were bred in-house and generated, as described previously [37], by crossing *TRPV1^Cre^* transgenic mice (B6.129-*Trpv1*^tm1[cre]Bbm^/J; Jackson Laboratory, stock number 017769; RRID: IMSR_JAX:017769) with *PC::G5-tdT* mice (B6;129S6-Polr2a^tm1(CAG-GCaMP5g,-tdTomato)Tvrd^/J; Jackson Laboratory, stock number 024477; RRID:IMSR_JAX:024477) to generate *TRPV1^Cre+/−^*;*GCaMP5g-tdT^f/+^* mice. Background control strains were used as wild-type (WT) mice. The mice were genotyped by Transnetyx, Inc. (Cordova, TN, USA) and were fed chow containing tamoxifen citrate (TD.140849, Envigo, 400 mg/kg) for 2 weeks to induce Cre activity, and switched to standard chow at least 1 week prior to experiments.

### 2.2. Chemicals and Reagents

Chemicals and reagents were purchased from Millipore Sigma (Burlington, MA, USA), unless otherwise stated.

### 2.3. Generation of Hnmt Flox Mice

The mouse *Hnmt* locus (ENSMUSG00000026986; Ensemble Project, EMBL-EBI, Hinxton, UK) was targeted by CRISPR-Cas9 to create a conditional knockout allele. A 5′ loxP site was inserted in intron 2, 131 bp upstream of exon 3, and a 3′ loxP site was inserted in intron 3, 171 bp downstream of exon 3. LoxP sites were introduced into mouse zygotes via electroporation to create targeted animals. Briefly, synthetic tracrRNA and crRNA [Integrated DNA Technologies (IDT), Inc., Coralville, IA, USA] were hybridized at an equimolar ratio by incubating at 95 °C for 5 min, followed by a ramp down to 25 °C. The crRNA/tracrRNA duplex was then assembled with Alt-R^®^ S.p. Cas9 nuclease protein (IDT Inc.) at 37 °C for 5 min to form RNP complexes. RNPs at a concentration of 50 ng/µL, along with 10 µM of ssODN template, were electroporated into C57BL/6N zygotes using a Gene Editor^TM^ electroporator (BEX Co., Tokyo, Japan) at 30 V, with 2 pulses of 1 ms with a 1 s interval, as previously described [38]. Edited embryos were implanted into pseudo-pregnant dams using standard approaches. The resulting founder litters were screened for edits as described below. The gRNA target, ssODN donor, and primer sequences are listed in Table 1.

Tail biopsy samples (<1 mm) were incubated overnight at 56 °C in a lysis buffer (0.1 mg/mL Proteinase K in 0.5% Triton X-100, 10 mM Tris pH 8.5), followed by heat inactivation at 85 °C for 15 min. For genomic DNA (gDNA), 1 μL of the tail lysis solution was used for subsequent PCR with Phire Green HSII PCR mastermix (F126L, Thermo Fisher, Waltham, MA, USA). Following genome editing, 5′-taget site, 3′-target site, and longer-range PCRs were performed with primers external to the homology arms of the ssODN templates. The resulting amplicons were purified and sent for Sanger sequencing [Azenta Life Science (formerly GENEWIZ) Inc., Burlington, MA, USA]. Sequencing was performed from gDNA isolated from founder G0 for initial screening, G1 animals, and G2 or later homozygous HNMT fl/fl to confirm the edited allele. Sequencing results confirmed that both 5′ and 3′ LoxP sites were intact and inserted at the intended target sites.

### 2.4. Whole-Mount Immunohistochemistry (IHC)

Colons from euthanized *Sox10^CreERT2^;Hnmt* mice and WT littermates were collected, fixed overnight in 4% paraformaldehyde for 2 h at room temperature, and processed for IHC, as described previously [39]. Antibody details are supplied in Table 2. Briefly, longitudinal muscle myenteric plexus (LMMP) preparations were rinsed three times (10 min each) in phosphate-buffered saline containing 0.1% Triton X-100 (PBST) followed by a 45 min incubation in blocking solution (containing 4% normal donkey serum, 0.4% Triton X-100 and 1% bovine serum albumin). Primary antibodies were diluted in blocking solution and applied overnight at room temperature. LMMPs were rinsed three times with PBS after removal of primary antibodies the following day, and secondary antibodies (diluted in blocking solution) were applied for 2 h at room temperature. Finally, LMMPs were rinsed in a 0.1 M phosphate buffer and mounted on slides with bicarbonate-buffered glycerol consisting of a 1:3 mixture of 142.8 mM sodium bicarbonate and 56.6 mM carbonate to glycerol. Images were acquired through the 20× (Plan-Apochromat, 0.8 n.a.) objective of a Zeiss LSM880 confocal microscope (Carl Zeiss, White Plains, NY, USA) with a 2× digital zoom.

### 2.5. Quantitative Real-Time Polymerase Chain Reaction (qRT-PCR)

Primers used for qRT-PCR were designed for the mouse *Hnmt* locus (ENSMUSG00000026986), are listed in Table 3, and were validated by confirmation of melting curve, R^2^, and efficiency with four standards by a dilution 1:10. Products were migrated on a 2% agarose gel with SYBR Safe (Thermo Fisher) and analyzed with Sanger Sequencing to confirm the specificity of the amplicons.

Total mRNA was isolated from distal colons with the NucleoSpin RNA/Protein Kit (740933, Macherey-Nagel, Allentown, PA, USA), treated with TurboDNAse (AM1907, Thermo Fisher), and converted to cDNA using the SuperScript III First Strand Synthesis SuperMix (11752050, Thermo Fisher). qPCR was performed on 8 ng RNA equivalent with a FastSYBR Green Master Mix (4385612, Thermo Fisher) using a 7500 Fast Real-time PCR System (Applied Biosystems, Foster City, CA, USA). Fold changes from WT littermates were calculated using the 2^−DDCt^ method [40]. Ribosomal protein S6 (RPS6) was used for normalization. Amplification products were migrated on 2% agarose gels and visualized with a GelDoc XRS+ (Bio-Rad, Hercules, CA, USA) to confirm the deletion of exon 3 of the *Hnmt* gene.

### 2.6. Visceral Hypersensitivity Measurements

Visceral sensitivity was measured in *Sox10^CreERT2^;Hnmt* mice and WT littermates by assessing visceromotor responses (VMRs) to colorectal distensions (CRDs) [41,42]. Intracolonic pressure was measured as previously described by a miniaturized pressure transducer catheter (SPR-524, Mikro-Tip catheter, Millar Instruments, Houston, TX, USA) equipped with a custom-made plastic balloon tied below the pressure sensor. Mice were trained to restrainers for 3 h one day prior to the experiment. Mice received an enema of vehicle (30% ethanol in saline) or histamine (1 mg/kg, dissolved in vehicle) [43] and, after 3 h, were briefly anesthetized with isoflurane and the pressure transducer balloon was inserted into the colorectum. Mice were then acclimated to restrainer for 30 min before starting the CRD procedure. Graded phasic distensions (20, 40, 60, and 80 mmHg, two times each, 20 s duration, 3 min interstimulus interval) were delivered to the balloon by a barostat and VMR recordings were acquired from the pressure transducer using PowerLabs 8.1.23 software (LabChart 8, AD Instruments, Colorado Springs, CO, USA). VMR recordings were performed as repeated measure experiments where the control recordings were performed one week prior to conducting experiments with histamine. Raw VMR traces were processed by running the SmoothSec and root mean square functions within the acquisition software to filter phasic colonic contractions [41]. Following filtering, the integral from the response at each distension pressure (20 s duration) and the baseline mean from 20 s prior to the distension were obtained. Responses were considered significant if they were at least 2 standard deviations above the baseline mean.

### 2.7. Ca^2+^ Imaging

Live whole mounts of the colonic myenteric plexus were prepared for Ca^2+^ imaging as previously described [44]. To evaluate the impact of glial HNMT on nociceptive signaling, *TRPV1^Cre^;GCaMP5g-tdT* preparations were treated with histamine dihydrochloride (10 µM; 3545, Bio-Techne, Minneapolis, MN, USA) for 30 min followed by a 3 μM capsaicin (360376, Millipore Sigma; 30 s bath application) challenge to stimulate TRPV1+ nociceptors. Images were acquired at 1 Hz through a 40× water immersion objective (LUMPlanFI, 0.8 numerical aperture) of an upright Olympus BX51WI fixed stage microscope (Olympus, Tokyo, Japan) using NIS-Elements software (v.4.5; Nikon, Tokyo, Japan) and an Andor Zyla sCMOS camera (Oxford Instruments, Abingdon, UK). Individual nerve fibers were identified according to tdTomato expression and morphology [45]. Whole mounts were continually superfused with 37 °C Krebs buffer consisting of (in mM): 121 NaCl, 5.9 KCl, 2.5 CaCl_2_, 1.2 MgCl_2_, 1.2 NaH_2_PO_4_, 10 HEPES, 21.2 NaHCO_3_, 1 pyruvic acid, and 8 glucose (pH adjusted to 7.4 with NaOH) at a flow rate of 2–3 mL min^−1^.

### 2.8. Data Analysis

Raw Ca^2+^ imaging files were analyzed with SparkAn 5.5.6.0 software (Adrian D. Boven and Mark T. Nelson, Department of Pharmacology, College of Medicine, University of Vermont, Burlington, VT, USA), where the regions of interest (ROIs) were drawn around individual nerve fibers identified in *TRPV1^Cre^;GCaMP5g-tdT* mice by tdTomato expression and morphology [45]. Analysis and generation of traces were performed using GraphPad Prism 9.5 (GraphPad Software, La Jolla, CA, USA) and NIS Elements Advanced Research 4.50 software (Nikon). The traces represent the change in fluorescence (ΔF/F) over time [46] for individual nerve fibers. The area under the curve (AUC) was calculated as (ΔF/F) per second. The time to response was calculated as the time from drug application to response. N values represent individual nerve fibers for Ca^2+^ imaging experiments.

Cell counts and ganglionic expression data were analyzed offline using FIJI 1.53t software (National Institutes of Health, Bethesda, MD, USA). The cell counts were performed using the cell counter plug-in of FIJI. Enteric neuron and glia numbers are presented as ganglionic packing density, which were calculated by tracing the ganglionic area and counting the number of HuC/D-immunoreactive neurons and S100b-immunoreactive glia, respectively, within the defined ganglionic area. Enteric glia and neuron histamine expression were measured by recording the mean grey value of histamine within HuC/D-immunoreactive neurons and S100b-immunoreactive glia, respectively. The relative ganglionic expression of glial fibrillary acidic protein (GFAP), S100b, H1R, and peripherin were measured by recording the mean gray values of GFAP, S100b, H1R, and peripherin within a defined ganglionic area. The cell counts and ganglionic expression data expressed as arbitrary fluorescence units (AFU) were performed on a minimum of 10 ganglia per animal and averaged to obtain a value for that animal. N values represent the number of animals in experiments.

Data were analyzed using Prism 9.5 and are shown as the mean ± SEM. The data were analyzed using a one-way or two-way analysis of variance (ANOVA) with a Bonferroni post-test where appropriate. A *p*-value less than 0.05 was considered significant.

## 3. Results

### 3.1. Cellular Distribution of HNMT in the Myenteric Plexus

We initially assessed potential cell types expressing HNMT by screening gene expression in published single-cell and bulk RNAseq datasets [37,47,48]. Based on gene expression, both enteric glia and neurons have the potential to express *Hnmt* in mice and humans; however, the levels of translating mRNA appear higher in glia than neurons (Figure 1A). To assess whether protein expression reflects gene expression patterns, we conducted immunofluorescence labeling with a commercial antibody that targets the N-terminal of the HNMT protein (Figure 1B–D). Positive labeling was observed within ganglia and intraganglionic connectives of the mouse colon myenteric plexus. Limited extraganglionic HNMT labeling was observed and could indicate expression by the intramuscular glia and/or macrophages. Within myenteric ganglia, HNMT co-localized with the majority of GFAP-positive enteric glia and the majority of peripherin-positive enteric neurons. Glial HNMT labeling was localized to cell bodies (Figure 1C; arrows) and processes of GFAP-positive glia. In neurons, HNMT labeling was present in the cell bodies (Figure 1C; asterisks) and varicose processes of peripherin-positive neurons. Therefore, the genetic and protein expression suggest that neuronal and glial HNMT contribute to histamine metabolism in the myenteric plexus.

### 3.2. Generation of a Cell-Type Specific Hnmt Ablation Model

Since both the neurons and glia express HNMT, we reasoned that cell-type specific strategies were necessary to understand the relative contributions of glia and neurons to histamine degradation in the myenteric plexus. Based on data from the central nervous system suggesting that astrocytes are the main site of functional HNMT expression, we focused on strategies to selectively perturb glial HNMT in the gut. To this end, we used CRISPR/Cas9 technology to create a conditional *Hnmt*-knockout model in which *Hnmt* can be ablated in a cell-dependent manner (Figure 2 and Appendix A). A founder with canonical loxP sites flanking exon 3 of the *Hnmt* gene was successfully targeted and the allele transmitted to G1 progeny (Figure 2A,B). Sequencing confirmed the correct insertion of intact LoxP sites into intron 2 and intron 3, respectively, and identified a 70 bp insertion of intronic sequence upstream of the 5′ loxP site in intron 2 (Appendix A).

To test the efficacy of the conditional *Hnmt* ablation model, floxed *Hnmt* mice were crossed with mice expressing Cre under the control of the broadly expressed cytomegalovirus (CMV) promoter (*CMV^Cre^;Hnmt* mice) to delete *Hnmt* in most cells and tissues, including the germ line [36]. Sequencing of the PCR amplicons amplified from gDNA confirmed Cre/LoxP recombination, resulting in the loss of exon 3 at the gDNA level in *CMV^Cre(+)^;Hnmt^f/f^* mice with a 401 bp deletion between the LoxP sites (Appendix A). Furthermore, the sequencing of cDNA transcribed from mRNA extracted from *CMV^Cre(+)^;Hnmt^f/f^* mice demonstrated that the loss of exon 3 results in the generation of a frameshifted transcript with the first premature stop codon occurring at amino acid position 47 (Figure 2C).

*Hnmt* ablation was also verified via qRT-PCR in the colon samples from WT, *CMV^Cre(−)^;Hnmt^f/∆^*, and *CMV^Cre(+)^;Hnmt^f/f^* mice (Figure 2 and Appendix A). The amplification of the exon 1 primer set, upstream of *Hnmt* excision, resulted in bands at 110 bp in male and female mice (Appendix A, respectively). The amplification of exon 3 primer set in male and female *CMV^Cre(+)^;Hnmt^f/f^* mice resulted in a faint or absent band at 246 bp (Appendix A, respectively). We also observed a shortened amplicon in male and female *CMV^Cre(+)^;Hnmt^f/f^* mice utilizing primer set exons 2–4 (Appendix A, respectively). Two bands were observed in *CMV^Cre(−)^;Hnmt^f/∆^* mice due to germline deletion of *Hnmt* (Appendix A). The quantification of *Hnmt* exon 1, exon 3, and exons 2–4 was normalized to *Rps6* (Figure 2D,E and Appendix A, respectively). *Hnmt* exon 1 was significantly decreased in *CMV^Cre(+)^;Hnmt^f/f^* mice (AU: 0.37 ± 0.06) compared to *CMV^Cre(−)^;Hnmt^f/∆^* and WT mice (AU: 1.50 ± 0.26 and 1.54 ± 0.23, *p* = 0.002 and *p* = 0.005, respectively; Figure 2D). *Hnmt* exon 3 was also significantly decreased in *CMV^Cre(+)^;Hnmt^f/f^* mice (AU: 0.25 ± 0.06) compared to *CMV^Cre(−)^;Hnmt^f/∆^* and WT mice (AU: 1.40 ± 0.16 and 1.64 ± 0.19, *p* < 0.0001 and *p* < 0.0001, respectively; Figure 2E). Furthermore, exons 2–4 were significantly decreased in *CMV^Cre(+)^;Hnmt^f/f^* mice (AU: 0.38 ± 0.04) compared to *CMV^Cre(−)^;Hnmt^f/∆^* and WT mice (AU: 1.41 ± 0.23 and 1.69 ± 0.27, *p* = 0.002 and *p* < 0.001, respectively; Appendix A). The decrease in expression of exon 3 deleted mRNA compared to controls indicates that the mutant transcript is likely degraded by the non-sense-mediated decay pathway—a mechanism that eliminates frameshifted mRNAs to prevent the translation of aberrant proteins [49].

The relative expression of exon 3 and exons 2–4 to exon 1 region was also assessed (Appendix A). Quantification via qRT-PCR showed a significant decrease in *Hnmt* exon-3-relative expression in *CMV^Cre(+)^;Hnmt^f/f^* mice (AU: 0.55 ± 0.08) compared to *CMV^Cre(−)^;Hnmt^f/∆^* mice and WT mice (AU: 1.05 ± 0.10 and 1.12 ± 0.11; *p* = 0.004 and *p* = 0.003, respectively; Appendix A). *Hnmt* exon-2–4-relative expression to exon 1 was unchanged in all groups, denoting that *Hnmt* excision was confined to exon 3 (Appendix A).

Following the validation of *Hnmt* ablation using a germ line-driven approach in *CMV^Cre^* mice, we tested the efficacy of a cell-specific approach by confining the *Hnmt* deletion to glia by crossing with *Sox10^CreERT2^* mice. Similar to *CMV^Cre^;Hnmt* mice, the deleted *Hnmt* allele was present in *Sox10^CreERT2^;Hnmt* mice following Cre induction by tamoxifen (Appendix A), but not in untreated Cre-expressing animals or Cre-negative controls. qPCR gene expression of *Dao* and *Nat2* were unchanged in the ileum of *Sox10^CreERT2^;Hnmt* mice, suggesting that compensatory mechanisms of histamine degradation do not occur following the ablation of glial *Hnmt* (Appendix A, respectively). Together, these data validate the efficacy of the *Hnmt flox* model to delete *Hnmt* and to do so in a cell-specific manner.

### 3.3. Effects of Glial Hnmt Ablation on Intracellular Histamine and ENS Structure

In prior work, we found that enteric glia and neurons contain histamine, and that changes in intracellular histamine in disease contexts can be detected via immunolabeling [34]. Based on this evidence, we hypothesized that deleting glial *Hnmt* would elevate intracellular histamine expression in glia and promote changes associated with reactive processes. Immunolabeling for histamine was again primarily localized to glia and a subpopulation of neurons in the myenteric plexus of WT littermates (Figure 3A,C). The overall cellular distribution of histamine immunoreactivity was similar in *Sox10^CreERT2^;Hnmt* mice; however, the quantification of immunolabeling data showed a greater intensity of labeling in S100b+ glia in male mice (Figure 3B,E; AU: 122.64 ± 13.48 vs. 177.08 ± 11.95, WT vs. *Sox10^CreERT2^;Hnmt*, respectively, *p* = 0.0497), with no difference observed in females (Figure 3D). Neuronal (HuC/D+) histamine labeling was unchanged in the *Sox10^CreERT2^;Hnmt* mice of both sexes (Figure 3F).

Prior work showed that increased glial histamine content is associated with changes in GFAP expression, indicating a potential transition to a “reactive”-type phenotype [34]. In support, GFAP immunofluorescence was altered in male *Sox10^CreERT2^;Hnmt* mice that exhibited stronger histamine immunoreactivity and was unchanged in females. However, GFAP expression was inversely correlated with histamine labeling and decreased in males (AU: 56.53 ± 3.84 vs. 37.35 ± 7.26, WT vs. *Sox10^CreERT2^;Hnmt*, respectively, *p* = 0.019; Figure 4A). In contrast, S100b immunofluorescence was positively correlated with histamine and increased S100b immunolabeling coincided with elevated histamine in glia from male *Sox10^CreERT2^;Hnmt* mice (AU: 125.87 ± 8.44 vs. 166.32 ± 3.43, WT vs. *Sox10^CreERT2^;Hnmt*, respectively, *p* = 0.021; Figure 4B). No changes in the neuron (HuC/D+) or glial (S100b+) cell density were detected in either male or female mice lacking glial *Hnmt* (Figure 4C,D). Therefore, changes in histamine degradation/content are not necessarily responsible for driving a change in the glial phenotype marked by increased GFAP expression and could impact the cell phenotype in a complex, sex-dependent manner.

### 3.4. Glial Hnmt Regulates Visceral Sensitivity in Male Mice

Histamine is considered an important mediator in processes that promote visceral hypersensitivity through actions on H1 receptors expressed by colon-innervating nociceptors [11,13]. Therefore, we speculated that limiting histamine degradation by deleting glial *Hnmt* would promote visceral hypersensitivity. In agreement with prior work [43], in vivo visceromotor responses (VMR) to colorectal distensions were potentiated by intracolonic histamine administered via an enema (Figure 5). Female mice appeared to have higher baseline sensitivity to colorectal distensions, particularly at higher distension pressures; however, female sensitivity was unaffected by intracolonic histamine. Interestingly, the sensitizing effect of histamine was confined to male mice, which increased the magnitude of VMR by 98% and 74% to vehicle (40 mmHg and 60 mmHg, respectively). In contrast to our predictions, deleting glial *Hnmt* protected against the ability of histamine to sensitize VMR instead of further potentiation.

To explain the results of the VMR experiments, we reasoned that elevated basal histamine levels resulting from the deletion of glial *Hnmt* could possibly desensitize or internalize nociceptor H1 receptors and cause insensitivity to histamine. H1 receptor immunolabeling and quantification were unaffected in myenteric ganglia from glial *Hnmt* knockout animals (Figure 6A–D and Figure 6E, respectively), which argues against a major role for changes in H1 receptor expression or localization in this model. Therefore, we focused on testing whether histamine-mediated nociceptor desensitization could explain the in vivo results in the glial *Hnmt* ablation model. To this end, we measured Ca^2+^ responses in Trpv1+ nociceptive nerve terminals innervating the colon myenteric plexus in *Trpv1^Cre^;GCaMP5g-tdT* mice following exposure to histamine (Figure 7).

Capsaicin-induced Ca^2+^ responses in the majority of Trpv1+ nerve fibers identified by tdTomato expression in male and female mice (Figure 7A and Figure 7C, respectively). Pre-incubation with histamine for 30 min did not alter the percentage of responding nerve fibers in either sex (Figure 7B,D) or the amplitude of Ca^2+^ response in males, but significantly decreased the amplitude of Ca^2+^ response in females by 58% (*p* < 0.0001, Figure 7I). Likewise, histamine decreased the AUC of Ca^2+^ responses induced by capsaicin in female nociceptor terminals by 64% (*p* < 0.01; Figure 7J). The time from drug application to initial response is a common indicator used to measure sensitivity in neurons. We assessed time to response after starting the capsaicin bath application as a measurement of Trpv1+ nerve fiber sensitivity, and found that prior exposure to histamine delayed responses by 29 ± 7 s in males and 13 ± 5 s in females (*p* < 0.01, Figure 7K). Together, these data support the conclusion that prolonged exposure to histamine, as would occur in male mice lacking glial *Hnmt*, can reduce the activity of nociceptive nerve terminals innervating the myenteric plexus and impair responses to colorectal distension.

## 4. Discussion

Histamine is an important neuromodulator in the gut and a mediator of neuroimmune interactions involved with neuroplasticity in DGBI. Mechanisms that regulate histamine availability surrounding enteric neurons and sensory nerve terminals are integral in this process, but poorly understood. Here, we show that enteric glia and neurons have the potential to contribute to histamine degradation in the ENS through their expression of the enzyme HNMT. We specifically assessed the relative contribution of glial HNMT and found that ablating glial *Hnmt* increases glial intracellular histamine levels and alters histamine-induced changes to visceral hypersensitivity to colorectal distention. Prolonged histamine exposure reduced Trpv1-mediated Ca^2+^ responses in nociceptive nerve terminals innervating the myenteric plexus. These effects were sexually dimorphic and could be involved in sex-biased mechanisms underlying visceral pain in DGBI, such as IBS.

Histamine exerts multiple effects on gut motor, secretory, and sensory functions by modulating the activity of enteric and DRG neurons. Multiple histamine receptor subtypes are involved and create heterogenous effects based on cellular and subcellular distribution. For example, histamine produces long-lasting excitation of enteric neurons in the submucosal and myenteric plexuses of the small and large intestine through effects on H2 receptors expressed on the cell bodies of enteric neurons [50,51]. H1 receptors also appear to play a role in the activation of human submucosal neurons by supernatants obtained from IBS mucosal biopsies [14]. This slow excitatory effect provides a long-lasting drive to maintain an immune “alarm” program that does not appear to desensitize over periods of multiple hours [52,53]. Such a program is thought to contribute to enhanced secretions and motility to expel potential pathogens from the gut lumen and can produce diarrhea associated with infections or food allergies. The microbiota is a major source of intestinal histamine [11] and it is possible that enteric neurons monitor, and control, bacterial species through a similar mechanism. In contrast to its excitatory effects on neuron cell bodies, histamine can also exert inhibitory effects on enteric neurotransmission through actions on presynaptic H3 receptors [54]. Here, histamine acts to suppress synaptic transmission at nicotinic synapses in enteric neurocircuits. Although the role of presynaptic inhibition by histamine is still poorly understood, it appears relevant for the effects produced by histamine release from mast cells [7,55,56]. Histamine release from degranulated mast cells also contributes to long-lasting excitation of submucosal and myenteric neurons and increases the sensitivity of sensory nerve fibers that project to the gut from dorsal root ganglia [57]. H1 receptors are involved in mechanisms that sensitize these nerves and H1 receptor antagonists improve symptoms and decrease visceral hypersensitivity in IBS patients [58] and animal models [59]. Histamine-driven sensitization of colon-projecting sensory nerves involves subsequent effects on Trp channels [12,13,43]; however, increased H1 and H2 receptor expression have also been reported in the mucosa of IBS patients [60]. Interestingly, incubating the myenteric plexus whole mounts with histamine for 30 min delayed Trpv1-mediated Ca^2+^ responses in nociceptive nerve terminals and reduced response magnitude in females. Whether these effects involve inhibitory presynaptic mechanisms similar to those present in enteric neurocircuits, desensitization of H1 receptors, or a disconnect between H1 receptors and Trp channels will require additional work. However, such a mechanism could explain the loss of histamine-induced visceral hypersensitivity observed here.

Changes in histamine production contribute to visceral hypersensitivity [12,13,40] and increased gut histamine is reported in IBS [10] and correlates with heightened pain sensations [11]. Greater quantities of extracellular histamine could result from either increased production or defects in degradation. Nearly all histamine clearance involves oxidation through the actions of DAO or methylation via HNMT. DAO is expressed in the intestinal epithelium [25] where it contributes to metabolizing extracellular histamine contained in food [9], and DAO deficiency contributes to histamine intolerance [26,27]. HNMT is the primary mechanism responsible for histamine degradation in the brain [28,29,30,31], and is mainly localized to astrocytes [32,33]. Deleting astrocytic *Hnmt* impairs clearance of evoked histamine release in the brain and disrupts motor coordination [61]. This contrasts the whole-body *Hnmt* ablation models which produce neurological and behavioral effects such as aggression and abnormal sleep [62]. These differences highlight the importance of understanding cell type-specific mechanisms of histamine clearance. Enteric glia express *Hnmt* [37,47] and take up histamine in the myenteric plexus [34]. Ablating glial *Hnmt* increased glial, but not neuronal, intracellular histamine, which supports the concept that enteric glia actively work to eliminate histamine in enteric neurocircuits. The benefit of understanding this mechanism in greater detail will likely reach beyond IBS pathophysiology, since changes in HNMT and its enzymatic activity have been linked to neurological disorders including Parkinson’s disease, multiple sclerosis, and schizophrenia [63,64,65].

Biological sex contributes to the etiology of DGBI, and females are often affected at a greater frequency and severity than males [66,67,68]. Sexual dimorphism is also observed in animal models of visceral pain, and the underlying mechanisms include differences in histamine production. For example, psychological stress increases histamine-containing granules in mast cells, mast cell histamine release, and serum histamine levels in female mice [69]. Most preclinical work assessing the role of histamine in visceral hypersensitivity has focused on the effects in male animals [7,12,43,70,71,72] without a clear assessment of potential sex differences. Our data support prior observations showing that intracolonic histamine sensitizes visceromotor responses in male animals [43]. Surprisingly, however, females were comparatively insensitive. Several observations could shed light on cellular mechanisms underlying these differences. First, differences in intracellular glial histamine content were only observed in males after *Hnmt* ablation. This is consistent with prior observations in the mouse neonatal maternal separation model, in which males exhibited increased ganglionic histamine levels accompanied by an increase in mast cells surrounding the myenteric plexus [34]. This observation could suggest sex-dependent differences in either the ability of glia to take up histamine or in the relative quantitates of extracellular histamine available to glia. Second, histamine appears to produce sex-dependent effects at the level of nociceptive nerve terminals in the myenteric plexus. Samples pre-incubated in histamine for an equivalent time as during the in vivo enema exhibited delayed Ca^2+^ responses mediated by Trpv1 in both sexes. However, histamine exposure had a greater affect in samples from females and also reduced response magnitude. The mechanisms underlying this sex difference in histamine on sensory transduction in the gut are unclear and will require additional work to be fully understood.

Histamine and histamine receptor antagonism have gained attention as potential therapeutic targets in DGBI such as IBS [73,74] and other chronic conditions [75], but the clinical outcomes of targeting these mechanisms are mixed, possibly because of an incomplete understanding of the mechanisms discussed above. Ketotifen, a mast cell stabilizer and H1 receptor antagonist, benefits a subset of IBS patients [58,76], as does disodium cromoglycate, a mast cell stabilizer, which reduces mast cell activation and improves symptoms in diarrhea-predominant IBS [77]. Other promising candidates include the H1 receptor antagonist ebastin, which improves abdominal pain and discomfort in IBS patient [13], is well tolerated with minimal side effects, and does not pass the blood–brain barrier [78]. H4 receptor antagonism improved clinical symptoms in animal models of DGBI [11,17,18], but remains untested in humans; however, JNJ-39758979, a H4 receptor antagonist, relieves symptoms from histamine-induced pruritus [75], while toreforant, another H4 receptor drug, had no benefits over placebo in eosinophilic asthma [79] and rheumatoid arthritis [80]. Indeed, manipulating histamine receptors or mast cell histamine production is not effective in all cases. The antihistamine, cyproheptadine, has beneficial effects on visceral hypersensitivity in children with DGBI; however, 32% of patients reported adverse effects [15]. Reducing mast cell infiltration with the anti-inflammatory drug mesalazine may also have beneficial effects [81], but not in all cases [82]. It is possible that an incomplete understanding of the mechanisms regulating visceral hypersensitivity by histamine contributes to limited efficacy with certain drugs. The results presented in this study support this conclusion and highlight the need to understand these mechanisms in a cell type- and sex-specific context.

## 5. Conclusions

Our findings delineate the cellular distribution of histamine degradation machinery in the mouse myenteric plexus and show that HNMT expression is distributed among enteric glia and neurons. We created and validated a new model to perturb *Hnmt* in a cell-specific manner, and found that deleting glial *Hnmt* increased intracellular histamine content and normalized histamine-induced visceral sensitivity in a sex-dependent manner. Elevated histamine exposure, as could occur in mice lacking glial *Hnmt*, reduced the activity of nociceptive nerve terminals innervating the myenteric plexus which could impair visceral sensitivity. Therefore, glial-mediated histamine metabolism via HNMT contributes to factors modifying neuroplasticity in the gut and may contribute to sex-dependent changes involved in the DGBI.

## Figures and Tables

**Figure 1 biomolecules-13-01651-f001:**
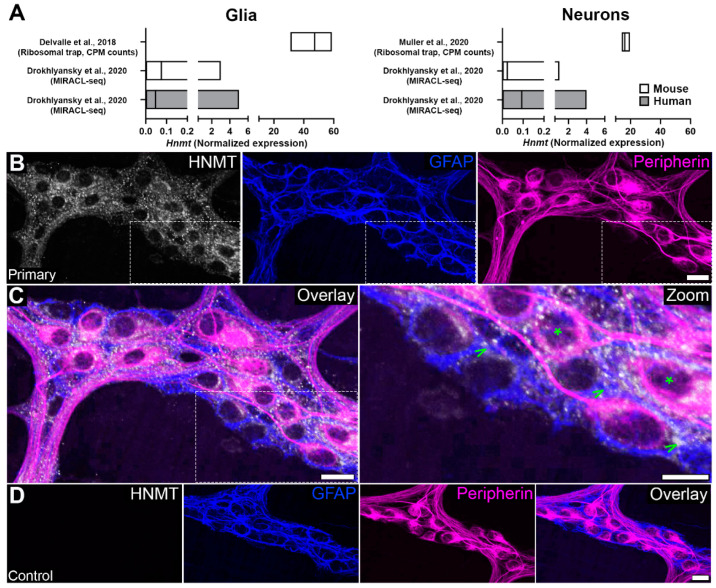
Histamine *N*-methyltransferase (HNMT) cellular distribution in the myenteric plexus. (**A**) *Hnmt* cell-specific transcript expression in combined glial and neuronal subpopulations (left and right, respectively) from the mouse and human colon prepared from published datasets. Graphs are prepared from the published datasets by [37,47,48] and data show normalized expression levels (mean ± range). (**B**–**D**) Confocal images showing HNMT immunoreactivity (grey), glial fibrillary acidic protein (GFAP) immunoreactivity (blue), and peripherin immunoreactivity (magenta) in the myenteric plexus of wild-type (WT) mice. (**B**,**C**) HNMT immunoreactivity is localized to the cell bodies and processes of the glia and neurons (**C**; arrowheads and asterisks (*) in zoom, respectively). (**D**) No primary control for HNMT antibody. Scale bars (**B**–**D**) = 20 µm. n = 3 animals.

**Figure 2 biomolecules-13-01651-f002:**
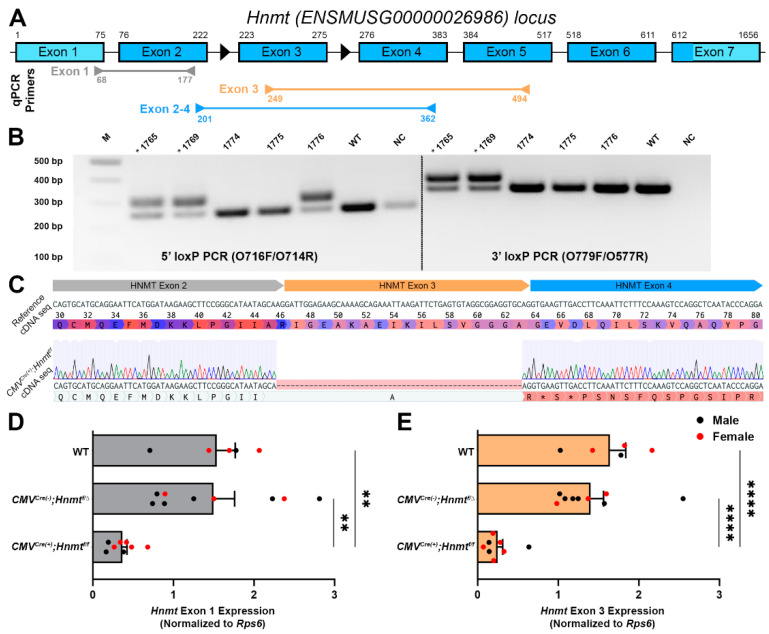
Histamine *N*-methyltransferase (*Hnmt*) *flox* targeting and validation. (**A**) Schematic of the targeted mouse *Hnmt* genomic locus with coding exons (blue), untranslated regions (light blue), loxP sites denoted by black arrowheads, and *Hnmt* exon map of primers for qPCR. The exon size and distance between the exons is not to scale. (**B**) Gel electrophoresis image showing PCR of 5′ and 3′ LoxP target regions of G1 progeny (n = 5) with wild-type (WT) and negative controls (NC). Asterisks (*) denote G1 animals containing an allele with both LoxP inserted. Top band—LoxP; bottom band—WT product. (**C**) Sequencing alignment of cDNA transcribed from RNA extracted from *CMV^Cre(+)^;Hnmt^f/f^* mice demonstrates successful Cre/LoxP recombination resulting in the deletion of exon 3 of the Hnmt cDNA. The Sanger chromatogram is aligned to reference WT cDNA sequence with the correct reading frame and translation annotated to show the resulting frameshift after A45 and the location of the first premature stop codon (*) at position 47 in the mutant sequence. (**D**,**E**) Quantification of qPCR data of relative expression of *Hnmt* exon 1 and exon 3 normalized to *Rps6* in colon samples from WT, *CMV^Cre(−)^;Hnmt^f/∆^*, and *CMV^Cre(+)^;Hnmt^f/f^* mice (**D** and **E**, respectively). n = 5–9 animals per group, ** *p* < 0.01, **** *p* < 0.0001, one-way ANOVA.

**Figure 3 biomolecules-13-01651-f003:**
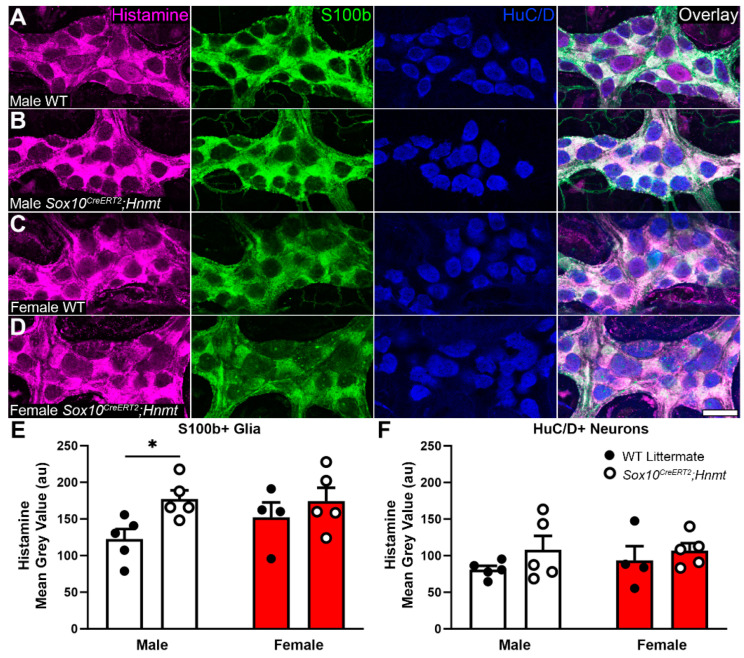
Glial histamine *N*-methyltransferase (*Hnmt*) ablation increases glial histamine in the colon myenteric plexus of male mice. (**A**–**D**) Confocal images showing histamine immunoreactivity (magenta), S100b immunoreactivity (glia, green), and HuC/D immunoreactivity (neurons, blue) in the myenteric plexus of WT and *Sox10^CreERT2^;Hnmt* (**A** and **C** and **B** and **D**, respectively) male and female mice (**A**,**B** and **C**,**D**, respectively). Scale bar in (**D**) = 40 µm and applies to (**A**–**D**). (**E**,**F**) The quantification of histamine in S100b+ immunoreactive glia and HuC/D+ immunoreactive neurons (**E** and **F**, respectively) in male and female WT and *Sox10^CreERT2^;Hnmt* mice. n = 4–5 animals per group, * *p* < 0.05, two-way ANOVA.

**Figure 4 biomolecules-13-01651-f004:**
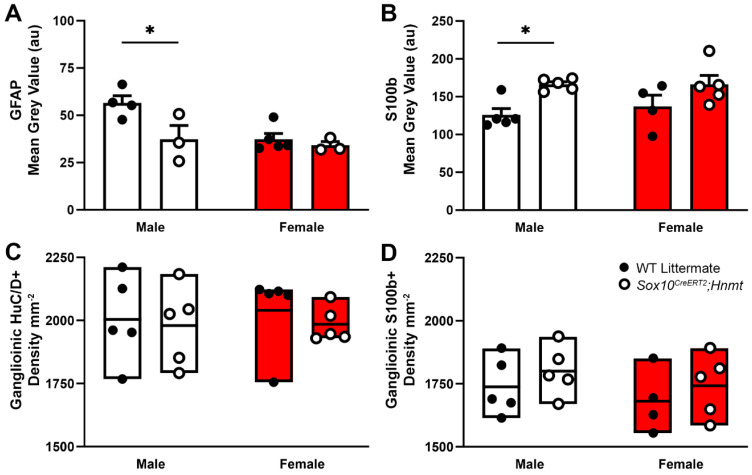
Effects of glial histamine *N*-methyltransferase (*Hnmt*) ablation on glial fibrillary acidic protein (GFAP) and S100b expression, and the ganglionic density of HuC/D+ neurons and S100b+ glia in the mouse colon myenteric plexus. (**A**–**D**) The quantification of ganglionic GFAP (**A**), glial S100b (**B**), ganglionic HuC/D+ neuron density (**C**), and ganglionic S100b+ glial density (**D**) in male and female WT and *Sox10^CreERT2^;Hnmt* mice. n = 3–5 animals per group, * *p* < 0.05, two-way ANOVA.

**Figure 5 biomolecules-13-01651-f005:**
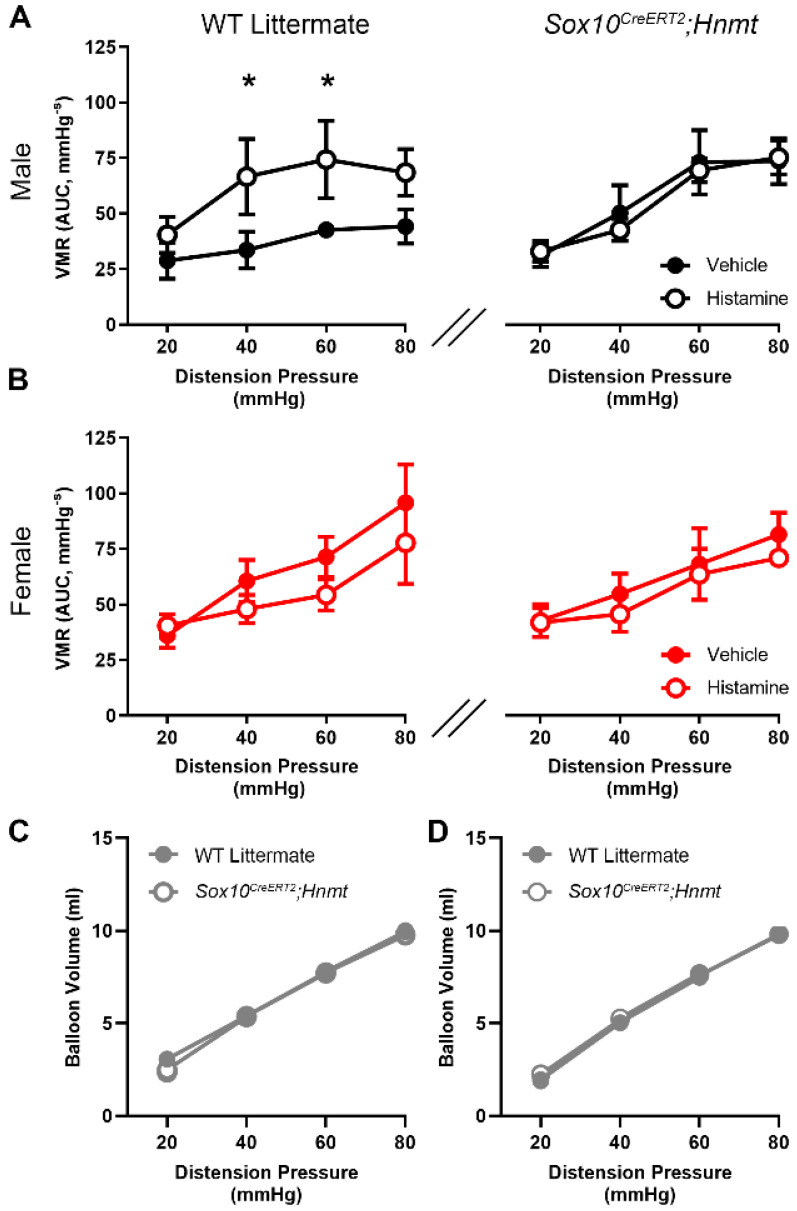
Effects of a histamine enema on visceromotor responses (VMRs) to colorectal distensions in *Sox10^CreERT2^;Hnmt* and WT littermates. (**A**) Histamine enema increased VMR in male control mice; however, this change was abolished in male *Sox10^CreERT2^;Hnmt* mice (left and right, respectively). (**B**) A histamine enema had no effect on VMR in female control or *Sox10^CreERT2^;Hnmt* mice (left and right, respectively). (**C**,**D**) Compliance measurements were equal between the WT littermates and *Sox10^CreERT2^;Hnmt* mice in males and females (**C** and **D**, respectively). n = 6–9 animals per group, * *p* < 0.05, 2-way ANOVA.

**Figure 6 biomolecules-13-01651-f006:**
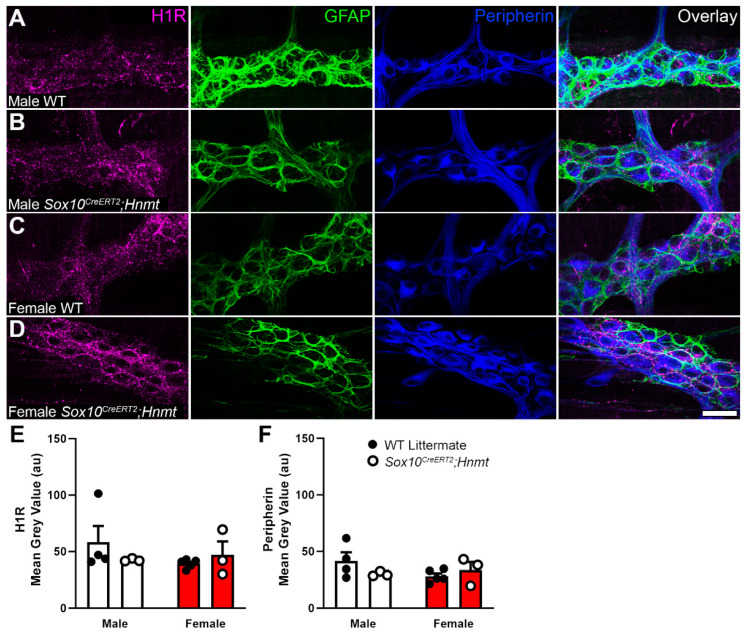
Effects of glial histamine *N*-methyltransferase (*Hnmt*) ablation on histamine type 1 receptor (H1R) and peripherin expression in the mouse colon myenteric plexus. (**A**–**D**) Confocal images showing H1R immunoreactivity (magenta), GFAP immunoreactivity (glia, green), and peripherin immunoreactivity (neurons, blue) in the myenteric plexus of WT and *Sox10^CreERT2^;Hnmt* mice (**A** and **C** and **B** and **D**, respectively) in males and females (**A**,**B** and **C**,**D**, respectively). Scale bar in (**D**) = 40 µm and applies to (**A**–**D**). (**E**,**F**) The quantification of H1R (**E**), and Peripherin (**F**) in male and female WT and *Sox10^CreERT2^;Hnmt* mice. n = 3–5 animals per group, two-way ANOVA.

**Figure 7 biomolecules-13-01651-f007:**
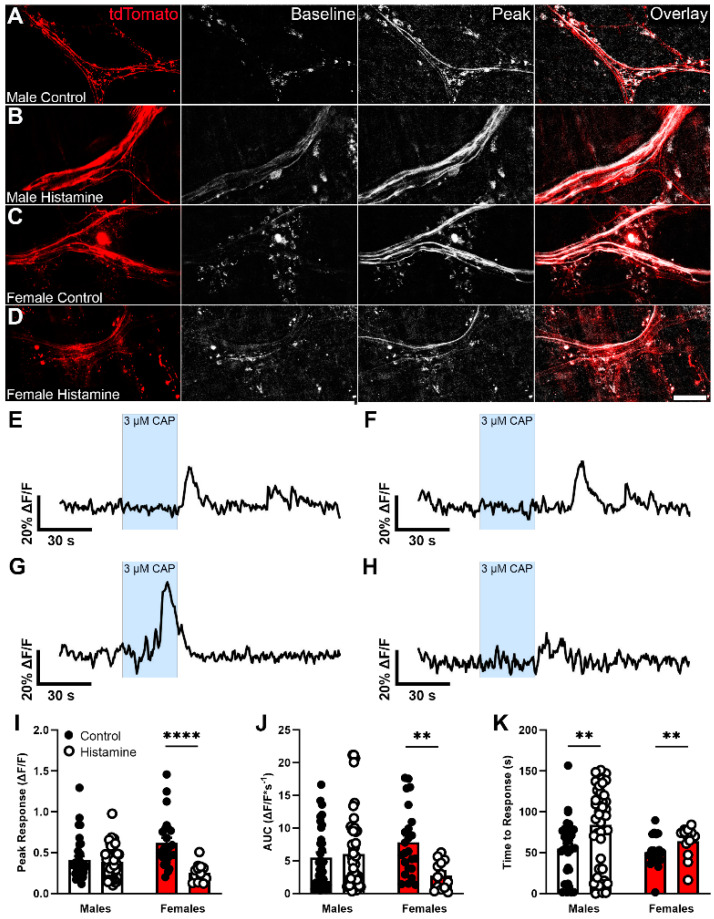
The effect of histamine on [Ca^2+^]_i_ responses in individual nerve fibers triggered by capsaicin (CAP) in *TRPV1^Cre^;GCaMP5g-tdT* mice. (**A**–**D**) Representative epifluorescence images showing tdT (red, *left*) expression and GCaMP5G fluorescence (grayscale, *middle 2*) in individual nerve fibers at the level of the myenteric plexus from male (**A**,**B**) and female (**C**,**D**) mice evoked by 3 µm CAP following exposure to control (**A**,**C**) or 10 µm histamine (**B**,**D**). (**E**–**H**) Representative traces showing [Ca^2+^]_i_ responses in individual tdT-positive nerve fibers evoked by 3 µm CAP in samples from male control, male histamine, female control, and female histamine (**A**, **B**, **C**, and **D**, respectively). (**I**–**K**) The quantification of the effects of histamine on the peak [Ca^2+^]_i_ response (**I**), the area under the curve (AUC, **J**), and the time to response (**K**) of nerve fibers. Scale bar in (**D**) = 60 µm and applies to (**A**–**D**). Data are representative of recordings in n = 13–47 individual nerve fibers from at least 3 mice. ** *p* < 0.01, **** *p* < 0.0001, 2-way ANOVA.

**Table 1 biomolecules-13-01651-t001:** gRNA target site, ssODN, and LoxP genotyping sequences.

Primer Set	Sequence
**5′gRNA (−)** **[5′-(N)20 NGG-3′]**	crRNA79: 5′-AAGAGGCCAACATGAAAGTG AGG-3′
5′ loxP ssODN donor template	O613: 5′-CTTCGCCTGTGTTTTCAGGCACTCTAATAAGAGAAACAGAAAT AAATAAAGAAGAGCCACAGACATGACCTCACGGATCCATAACTTCGTATA GCATACATTATACGAAGTTATGAATTCTTTCATGTTGGCCTCTTGTCCATTCATC CATTCTCTTTTGCTTATCAATAGACTTGATAGGAAGAATTGGC-3′
5′ LoxP genotyping primers	O716 Fwd: 5′-GCCTGTGTTTTCAGGCACTC-3′ O714 Rev: 5′-CACCTCCGCCTACACTCAGA-3′
**3′ gRNA (−)** **[5′-(N)20 NGG-3′]**	crRNA80: 5′-TAGTGAAGGCTAAAGTCCCAAGG-3′
3′ loxP ssODN donor template	O614: 5′-GCCACTTTACTTTTGTAAAATATCAAGTTATAAATACTAGCAAC ATCATTCAACTCAAATTCATTGCTCTCAATATATGTATAGGCCTTGGGAATTCA TAACTTCGTATAGCATACATTATACGAAGTTATAAGCTTGACTTTAGCCTTC ACTATAATTTGTAGGATAGGTAGAACTGAGAA-3′
3′ LoxP genotyping primers	O779 Fwd: 5′-AGATTCTGAGTGTAGGCGGA-3′ O577 Rev: 5′-TTTCCTTCCCTCACATGGGC-3′

**Table 2 biomolecules-13-01651-t002:** Primary and secondary antibodies used in this study.

Antibody	Source	Catalog No.	Resource ID	Dilution
* **Primary Antibodies** *				
chicken anti-GFAP	Abcam, Cambridge, MA, USA	ab4674	AB_304558	1:1000
chicken anti-S100b	Synaptic Systems, Göttingen, Germany	287 006	AB_2713986	1:1000
rabbit anti-H1R	Alomone, Jerusalem, Israel	AHR-001	AB_2039915	1:200
rabbit anti-Histamine	Millipore Sigma, Burlington, MA, USA	H7403	AB_260077	1:200
rabbit anti-HNMT	Atlas Antibodies, Bromma, Sweden	HPA035481	AB_10672094	1:400
biotin mouse anti-HuC/D	Abcam, Cambridge, MA, USA	A21272	AB_2535822	1:200
mouse anti-Peripherin	Santa Cruz, Dallas, TX, USA	sc-377093	AB_2923264	1:100
* **Secondary Antibodies** *				
goat anti-chicken Dylight 405	Jackson Labs, West Grove, PA, USA	103-475-155	AB_2337389	1:400
donkey anti-rabbit Alexa Fluor 488	Jackson Labs, West Grove, PA, USA	711-545-152	AB_2313584	1:400
goat anti-mouse IgG2a Alexa Fluor 488	Jackson Labs, West Grove, PA, USA	115-545-206	AB_2338855	1:400
goat anti-mouse Alexa Fluor 594	Jackson Labs, West Grove, PA, USA	115-585-207	AB_2338887	1:400
donkey anti-rabbit Alexa Fluor 647	Jackson Labs, West Grove, PA, USA	711-605-152	AB_2492288	1:400

GFAP, glial fibrillary acidic protein; HNMT, histamine *N*-methyltransferase; H1R, histamine type 1 receptor.

**Table 3 biomolecules-13-01651-t003:** Primer sequences for qRT-PCR used in this study.

Primer Set	Forward 3′-5′	Reverse 5′-3′
*mRPS6*	GAAGCGCAAGTCTGTTCGTG	GTCCTGGGCTTCTTACCTTCT
*mDao*	ATCTGCTGTGACGACTCCTC	TGGCCAAAGTCAGATTCTTGG
*mHnmt* Ex1	AGCTGCTGAGAACCCAATATG	CACTGGTGTTCCGTGGAATTA
*mHnmt* Ex2–4	AGCTTCCGGGCATAATAGCAA	CAGCACTTGGCTCAACAACT
*mHnmt* Ex3	AGATTCTGAGTGTAGGCGGAG	ACTTCGGTGGCTCTTCTTCT
*mNat2*	GGATGGTGTCTCCAGGTTAAT	CATGCCACTGCTGTACTTATT

*Dao*, diamine oxidase; *Hnmt*, histamine *N*-methyltransferase; *Nat2*, N-acetyltransferase 2.

## Data Availability

The data from this study are available from the corresponding author upon reasonable request.

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
