# Peer review of "Sexually Dimorphic Effects of Histamine Degradation by Enteric Glial Histamine *N*-Methyltransferase (HNMT) on Visceral Hypersensitivity"

_biomolecules, 2023, doi:10.3390/biom13111651_

Round 1
Reviewer 1 Report
Comments and Suggestions for Authors
The study by McClain and colleagues investigates the effects of histamine degradation by enteric glial hnmt on visceral hypersensitivity. The authors have previously shown that enteric glia detects extracellular histamine and they now hypothesize that histamine metabolism by enteric glia via hnmt might regulate visceral hypersensitivity. The authors test the hypothesis by creating a conditional knock-out of Hnmt in enteric glia and observe that ablation of hnmt in glia results in changes in content of intracellular histamine and visceral hypesensitivity in a sex-dependent manner.
Overall, the study is well written and is mostly easy to follow. The topic is of interest also for its possible application to DGBI. However, few issues should be clarified.
-The authors say that histamine is localized in glia and a subpopulation of neurons in the myenteric plexus of WT mice and that when Hnmt is ablated histamine concentrate more in S100b + glia than neurons in male mice and not females. It is unclear which subpopulation of neurons the authors are talking about. In addition it would seem that Figure 3 is problematic. Figure 3A-D:
While the pictures are beautiful, the specificity of histamine staining, particularly in panels C and D seems low, as there seem to be a diffuse staining everywhere. While there is no significance, when looking at the overlay pictures for C and D seem to indicate that there is more overall histamine in enteric glia in females, unless there is some non-specific staining for these slides. Color saturation of the overlay panel should also be controlled. In the overlay the green of GFAP is not green but almost white.
In panel F there would seem to be the same amount of histamine in neurons in WT and Sox10CreERT2;Hnmt mice and the data is homogenous, but looking at the overlay in panel A, it would appear as if WT HuC/D+ neurons contain more histamine than the Sox10CreERT2;Hnmt HuC/D+ neurons in males and perhaps also in females (even though in females is difficult to understand because of the histamine staining). If that is a representative picture of the entire group, then it would seem that values should be higher in WT HuC/D+ neurons than Sox10CreERT2;Hnmt HuC/D+ neurons, meaning that deleting glial Hnmt shifts the histamine expression to glia (still in agreement with the authors statement).
-In the picture for S100b in panel 3B is not very evident that S100b is increased in males, perhaps another picture would show it better?
-Representative images of GFAP expression should be added to Figure 4, as well as images showing co-staining of histamine and GFAP (similar to figure 6 and 3).
-Once the authors have shown that in male mice S100b+ enteric glia stores more histamine, they perform colorectal distension (CRD) experiments in WT and Hnmt-ablated mice using a histamine enema to potentiate the VMRs to CRD. They found that histamine only increaed VMRs to CRD in male WT mice and not in male with Hnmt ablated from enteric glia or WT female. Interestingly, the VMRs to CRD of females appear to be much higher in WT mice (almost double in females than males looking at the vehicle trace) and just a little bit higher in the hnmt glia-ablated female mice. The authors do not discuss this finding but perhaps should mention it. Might this hyperesponsiveness of females affect responses to the histamine enema?
-The authors also speculate that this loss of response to histamine in hnmt-ablated male mice might be the effect of having an increase in histamine, that was not degraded by enteric glia, which would desensitize histamine receptors, and in particular H1R. However, they found no differences in H1 receptor among groups of mice.
-In Figure 6 it would seem that female mice without Hnmt present with more glia+ for H1R staining (panel D), as it would look as almost all H1R staining overalps perfectly with the enteric glia net. Did the authors quantify this double staining as they did for S100b+ glia positive for histamine in Figure 3? Did the authors also co-stain with S100b besides GFAP, as other findings showed significance also in S100b+ glia? In addition, Figure 6 lists a panel G that is missing.
-The authors then looked into the possibility that an increase in histamine would desensitize nociceptors. To test this they measured Ca2+ responses in TRPV1+ nociceptive nerve terminals innervating the colon myenteric plexus following exposure to histamine. Here they observe that only female nerve fibers had a decreased amplitude of response and a reduced response to capsaicin with histamine pre-incubation. Both males and females presented with delayed responses after histamine pre-incubation, with males having the longer delay. Thus, they conclude that prolonged exposure to histamine is responsible for the altered VM responses observed. I would add "in males", because remains unclear what is happening in females which don't appear to have an increase in histamine, at least where the authors looked (glia, neurons) (Figure 3). Many questions remain unanswered. Do females have more histamine in general? Following the hypothesis of the authors, this would explain why WT female mice are already non-responsive to histamine enema. Is hnmt expressed more in other cell types in females? Explaining perhaps why hnmt in glial cells is not playing the same role in female mice as in male mice?
-The role of microbiome should be considered. Adding 16S gene analysis of WT and Hnmt-glia-ablated male and female mice might greatly add to the story, and perhaps shed some light on some of the more difficult to interpret results.
Author Response
Please see point-by-point response attached

Reviewer 2 Report
Comments and Suggestions for Authors
1. I concern about the inconsistency in the number of animals in the two compared groups in Figures 2, 3, 4, 6 and 7, which maybe affect the statistical results.
2. It is recommended that 2.7 should be moved forward to 2.2, so that 2.1 and 2.2 are materials and the rest are methods, meanwhile, it's best to combine 2.8 and 2.9.
Author Response

(The authors gave the same response as above.)

Round 2
Reviewer 1 Report
Comments and Suggestions for Authors
I recognize the author's attempt to respond to my comments within the given time of 5 days. While the authors confirm to have changed panels in Figure 3 and 6, the figures are exactly the same (at least to my eye) as the ones in the previous version. I understand that 5 days are such a short time that it is difficult to address all comments.